# The structure of the bacterial DNA segregation ATPase filament reveals the conformational plasticity of ParA upon DNA binding

Alexandra V. Parker[1], Daniel Mann [1,4], Svetomir B. Tzokov [1], Ling C. Hwang [1,3✉] & Julien R. C. Bergeron [1,2✉]

The efficient segregation of replicated genetic material is an essential step for cell division. Bacterial cells use several evolutionarily-distinct genome segregation systems, the most common of which is the type I Par system. It consists of an adapter protein, ParB, that binds to the DNA cargo via interaction with the *parS* DNA sequence; and an ATPase, ParA, that binds nonspecific DNA and mediates cargo transport. However, the molecular details of how this system functions are not well understood. Here, we report the cryo-EM structure of the *Vibrio cholerae* ParA2 filament bound to DNA, as well as the crystal structures of this protein in various nucleotide states. These structures show that ParA forms a left-handed filament on DNA, stabilized by nucleotide binding, and that ParA undergoes profound structural rearrangements upon DNA binding and filament assembly. Collectively, our data suggest the structural basis for ParA's cooperative binding to DNA and the formation of high ParA density regions on the nucleoid.

[1] Department of Molecular Biology and Biotechnology, the University of Sheffield, Sheffield, UK. [2] Randall Centre of Cell and Molecular Biophysics, King's College London, London, UK. [3] School of medicine, Faculty of Health, Education, Medicine and Social Care, Anglia Ruskin University, Chelmsford, UK. [4] Present address: Ernst-Ruska centre 3, Forschungszentrum Jülich, Jülich, Germany. ✉email: ling.hwang@aru.ac.uk; julien.bergeron@kcl.ac.uk

DNA replication and segregation are essential in all life forms. In bacteria, partitioning (Par) systems are responsible for the efficient segregation of replicated genetic material (chromosome, low-copy-number plasmids) to daughter cells[1–4]. The *par* loci are divided into three different types, classified by their NTPase protein: type I *par* loci encode a P loop ATPase, ParA, which possesses a deviant Walker-type motif; the type II ATPase, ParM, is actin-like; and the type III GTPase, TubZ, is tubulin-like. The mechanisms of type II and III systems have been extensively characterized[4,5]. In contrast, the mechanism of the type I segregation system remains elusive, with in particular discrepancy regarding ParA's action during segregation[6], despite being found in ~70% of bacteria.

The type I segregation system locus encodes the ATPase ParA; an adapter protein, ParB; and contains a centromere-like *parS* site(s). ParB binds preferentially to its cognate *parS* site[7] although it also possesses sequence-independent DNA-binding activity[8–11], and was shown recently to have CTPase activity[12–14], although the role of this activity remains unclear. ParA also binds to DNA in the presence of nucleotide, and this interaction is sequence independent. Biochemical studies have shown that ParB stimulates ParA's ATPase activity, promoting its dissociation from DNA[15–18], via a conserved arginine finger-like motif[19].

Several mechanistic models have been proposed for the type I segregation system. A mitotic-like filament model was initially suggested[20–23], similarly to type II and type III segregation systems. According to this model, ParA forms filaments in the presence of ATP, and filament dissociation upon interaction with ParB causes a pulling of the chromosome. However, a range of evidence, including the lack of observation of ParA filaments in vivo, has led to questioning of this model[24].

More recently, an alternative diffusion ratchet model has been proposed[25], whereby the increase in ATPase activity of ParA upon binding to ParB causes its dissociation from the DNA and an uneven distribution of ParA across the nucleoid; the partition complex then chases the ParA concentration gradient across the nucleoid whilst under confinement by the inner membrane, preventing diffusion of the partition complex into the cytosol. This model is supported by recent evidence using reconstituted systems and single-molecule measurements[9,17,26]; however, the molecular details of how it allows the diffusion of entire DNA molecules across the bacterial cell is currently not understood.

Type I segregation systems can be subdivided into two families, Ia and Ib, based on the ParA sequence[27], with type Ia ParA proteins possessing an additional N-terminal helix-turn-helix domain (NTD) that is involved in site-specific DNA binding for *par* gene transcription repression[18,28]. Type Ia systems are found in low-copy number plasmids, whereas type Ib systems are predominantly present in bacterial chromosomes. The organization of the locus also differs between these two subtypes: in type Ia, the *parA* gene is located directly after the promoter sequence, followed by *parB* and *parS*. In contrast, in type Ib systems, *parS* is often located after the promoter, followed by *parA* and *parB*, although in many bacterial species, multiple *parS* sites are dispersed throughout the chromosome[29].

The ParA superfamily is highly divergent, with low sequence identity between homologues. Nonetheless, core conserved regions are vital amongst ParAs, particularly the dimerization and nucleotide binding and DNA-binding sites[30]. The crystal structure of ParA has been solved in a range of bacterial species and plasmids[5,31–33], which revealed that despite low sequence identity the overall structure is conserved, and that they form dimers along the same interface. Negative-stain electron microscopy of several ParA orthologues, both of the type Ia and type Ib families, have shown the formation of filaments in the presence of nucleotide and/or DNA[22,32–34]; however, crystal structures of

ParA proteins bound to DNA did not provide any support for filamentous architecture[32]. Whether ParA proteins form filaments, and the molecular basis for filament assembly, remains controversial.

*Vibrio cholerae* is a Gram-negative bacterium, and the aetiological agent of cholera, a severe diarrheal disease affecting an estimated 3–5 million worldwide[35]. *V. cholerae* possesses two chromosomes: chromosome 1 (Chr1) and chromosome 2 (Chr2), that are ~3 Mbp and ~1 Mbp, respectively[36]. Each chromosome encodes its own segregation complex, with Chr1 encoding a chromosomal type Ib system, and Chr2 encoding a plasmid-like type Ia system (Fig. 1a). During cell division, both chromosomes segregate in synchronization. Chr1 initiates segregation first, from the old cell pole to new in an asymmetric manner. Once Chr1 reaches the mid-cell region, Chr2 commences segregation. Chr2 segregates symmetrically moving from the mid-cell to quarter cell positions, both chromosomes terminating segregation in unison[37–39].

Recent studies on the *V. cholerae* chromosome 2 ParA ($ParA2_{vc}$) have shown that it is a weak ATPase, and binds non-specifically to DNA[34,40], similar to other ParA orthologues[6,10,18]. It also revealed that it forms higher-order assemblies in the presence of DNA, with negative-stain EM analysis confirming the formation of filaments. $ParA2_{vc}$ binds ATP, leading to a slow conformational change to a DNA-binding active state. This then licenses $ParA2_{vc}$-ATP dimers to cooperatively bind onto DNA to form higher order complexes. $ParB2_{vc}$ stimulates $ParA2_{vc}$'s ATPase activity, leading to its dissociation from DNA[40]. This fast $ParA2_{vc}$ disassembly from the partition complex, coupled with rate-limiting nucleotide exchange, was proposed to generate dynamic $ParA2_{vc}$ gradients in *V. cholerae* cells. However, it is not known how $ParA2_{vc}$ forms higher-order assemblies on DNA, what are the structural changes of $ParA2_{vc}$ dimers upon DNA binding and the role of these filaments in $ParA2_{vc}$ dynamic gradients.

In this work, we report the structure of the $ParA2_{vc}$ filament bound to DNA, determined by cryo-EM. This structure reveals an unexpected set of contacts along the length of the dimer, and onto the DNA. We also present the crystal structures of $ParA2_{vc}$ in the apo and nucleotide-bound states. Collectively, these structures reveal a remarkable remodelling of the $ParA2_{vc}$ dimer upon filament formation, providing a structural basis for the cooperativity of its DNA binding, and suggest a molecular mechanism for type I segregation systems.

## Results

**Crystal structure of $ParA2_{vc}$.** $ParA2_{vc}$ has low sequence similarity to other ParA homologues, the *E. coli* P1/P7 plasmid ParAs being the closest orthologue of known structure (29% identity). We therefore sought to characterize the $ParA2_{vc}$ structure, to verify that it adopts a similar architecture to other ParA proteins, and to identify any differences with other ParA orthologues.

To this end, we purified $ParA2_{vc}$, and used negative-stain EM to image $ParA2_{vc}$ particles, in a range of nucleotide states. As shown in Supplementary Fig. 1, 2D classification of these particles shows the presence of a 2-lobed, V-shaped structure, consistent in shape and dimensions to the P1 ParA dimer reported previously[31]. This suggests that $ParA2_{vc}$ also forms dimers, in all the nucleotide states, as well as in the absence of nucleotides, as we also observed by SEC-MALS[40]. It is noteworthy that we did not observe any higher-order oligomerization/filament formation in any nucleotide state, unlike that reported in some other ParA orthologues[22,23], but similar to a previous study on $ParA2_{vc}$[34].

We next sought to determine the structure of the $ParA2_{vc}$ dimer. The purified protein crystallized readily, and the obtained

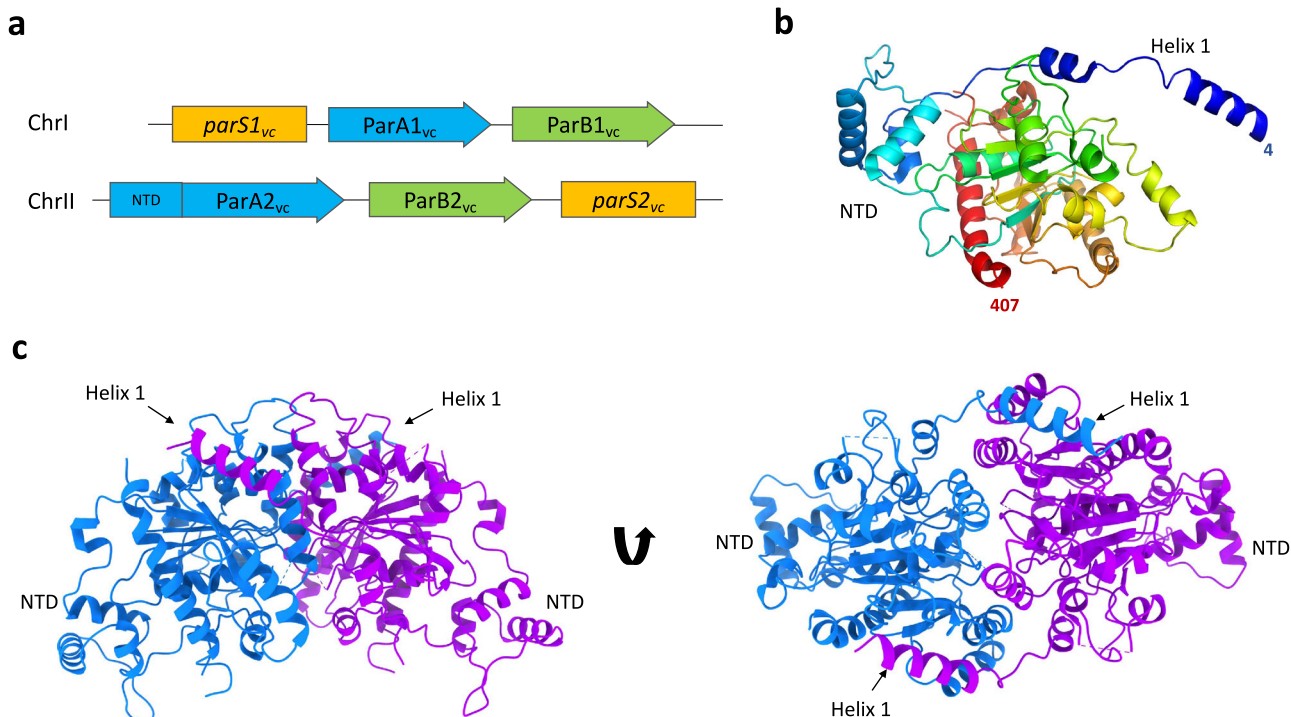

**Fig. 1 Structure of the ParA2$_{vc}$ dimer. a** Schematic representation of the chromosome segregation systems for the *V. cholerae* chromosome 1 (top), and chromosome II (bottom). The centromere-like partition site is in yellow *(parS)*, the adapter *parS* binding protein is in green (ParB) and the NTPase protein is in blue (ParA), with the additional N-terminal domain (NTD) found only in ChrII shown. **b** Cartoon representation of the ParA2$_{vc}$ crystal structure, in rainbow coloring, starting from blue at the N terminus, to red at the C terminus. **c** The crystallographic dimer of ParA2$_{vc}$ shown from the side (left) and top (right), with the two symmetry-related pairs in blue and magenta, respectively. The NTD and helix 1 are indicated.

crystals diffracted to ~2.5 Å. We were able to solve the structure by molecular replacement using the P7 ParA crystal structure as a template (see Materials and Methods for details), which allowed us to build an atomic model (Table 1).

The obtained ParA2$_{vc}$ crystal structure includes one ParA2$_{vc}$ molecule per asymmetric unit (Fig. 1b). The overall structure of the ParA2$_{vc}$ monomer is similar to that of P7 ParA, with an overall RMSD of 2.2 Å for CA atoms. In particular, the N-terminal HTH domain resembles that of P1 and P7 ParA, confirming that ParA2$_{vc}$ belongs to the type Ia family (Fig. 1b, Supplementary Fig. 2). We do nonetheless note some significant differences with these structures, most significantly in the position of the N-terminal helix (helix 1), present in both orthologues, but whose position differs significantly (Supplementary Fig. 2). No ligand density was observed in the active site, confirming that this structure corresponds to the apo state of the protein. It is worth noting that the nucleotide-binding site is generally poorly resolved, and in particular we were not able to build the P-loop in this structure.

As indicated above, our EM analysis suggests that ParA2$_{vc}$ is able to form dimers in the absence of nucleotide, at least at high concentration. We therefore questioned if crystallographic symmetry-related ParA2$_{vc}$ molecule pairs might recapitulate the biological dimer. As shown in Fig. 1c, one of the symmetry-related pairs is consistent with a biological dimer, and largely resembles the P7 ParA dimer structure (Supplementary Fig. 2). This suggests that we have obtained the structure of the ParA2$_{vc}$ dimer through crystallographic symmetry. Comparing the ParA2$_{vc}$ dimer to that of the P7 ParA structure (Supplementary Fig. 2) confirms that they possess a similar architecture, with helix 1 forming a domain-swapped interaction with the adjacent

**Table 1 X-ray crystallography data collection and refinement statistics.**

|  | apo | ADP-bound |
|---|---|---|
| *Data collection* |  |  |
| Space group | P 3$_2$ 1 2 | P 6$_1$ 2 2 |
| Cell dimensions |  |  |
| *a, b, c* (Å) | 63.247 63.247 | 199.205 199.205 |
|  | 214.373 | 260.066 |
| *α, β, γ* (°) | 90 90 120 | 90 90 120 |
| Resolution (Å)* | 2.601 (2.694 - 2.601) | 3.184 (3.298 - 3.184) |
| $R_{merge}$* | 0.01654 (0.3752) | 0.02072 (0.5636) |
| *I* / σ*I** | 36.81 (1.85) | 21.69 (0.98) |
| Completeness (%)* | 91.02 (44.89) | 96.38 (66.68) |
| Redundancy* | 2.0 (2.0) | 2.0 (2.0) |
| *Refinement* |  |  |
| Resolution (Å) | 2.6 | 3.2 |
| No. of reflections | 28240 | 99408 |
| $R_{work}$/$R_{free}$ | 0.2743/0.3353 | 0.2380/0.2779 |
| No. of atoms |  |  |
| Protein | 2899 | 11941 |
| Ligand/ion | 0 | 188 |
| Water | 47 | 0 |
| *B-factors* |  |  |
| Protein | 85.4 | 124.12 |
| Ligand/ion | / | 124.56 |
| Water | 64.4 | / |
| R.m.s. deviations |  |  |
| Bond lengths (Å) | 0.017 | 0.0031 |
| Bond angles (°) | 2.33 | 0.780 |

*Values in parentheses are for highest-resolution shell.

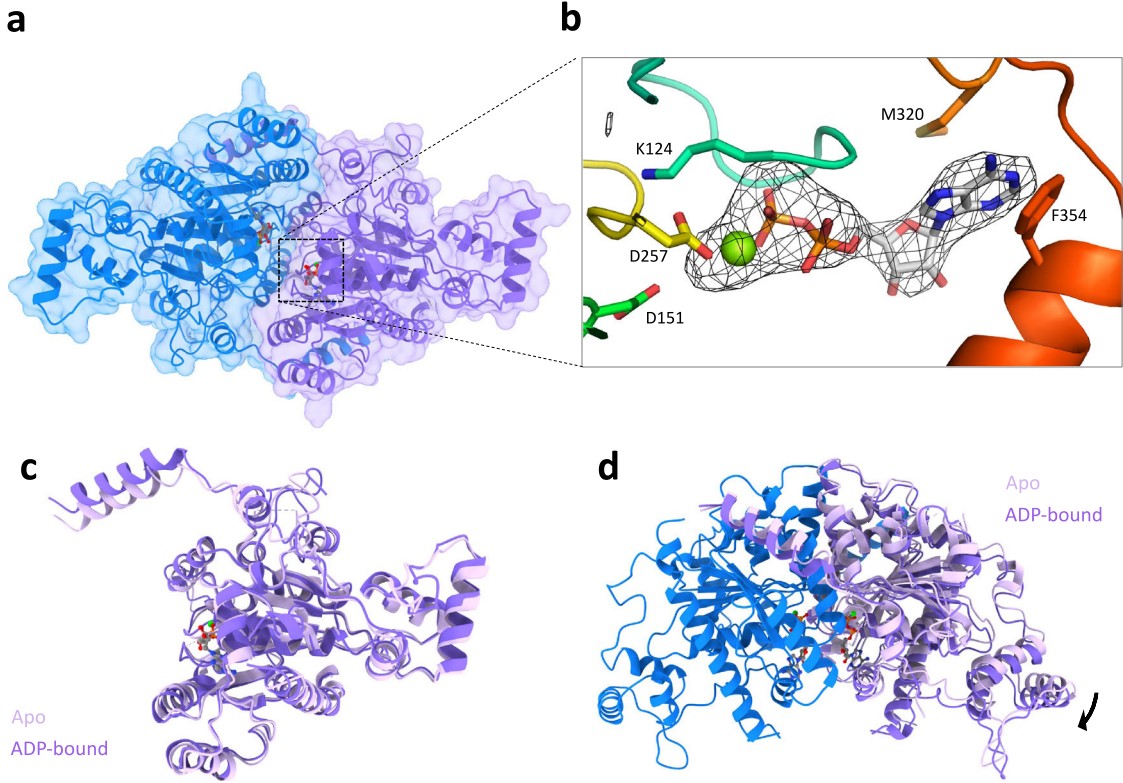

**Fig. 2 Structure of ParA2_vc bound to ADP. a** Surface representation of the crystal structure of a ParA2_vc-ADP dimer, coloured as in Fig. 1c. The ADP and Mg molecules, present in each ParA2_vc molecules are shown in sticks and sphere representations, respectively. **b** Close-up view of the ADP and Mg in one of the ParA2 molecules. The composite omit map is shown around the nucleotide. Residues that interact with the nucleotide and Mg are shown in sticks. **c, d** Overlay of the ParA2vc structure in the apo state (light pink) and ADP-bound state (dark pink). A monomer of each structure is shown in **c**, and the dimer is shown in **d**, overlaid on the blue subunit; only one chain of the apo structure is shown for clarity.

subunit. As indicated above, there is a difference in the position of helix 1, and as a consequence, the relative orientation of the dimer is slightly shifted in the ParA2_vc dimer compared to that in the P7 ParA dimer (Supplementary Fig. 2).

Overall, this structure confirms the common architecture of ParA proteins, and is consistent with the fact that the *V. cholerae* chromosome 2 is plasmid-like, as suggested previously[41], since the ParA2_vc structure confirms that it belongs to the type Ia family.

**Structure of ParA2_vc bound to nucleotide.** Previous studies on the type Ia P1/P7 ParA suggested structural rearrangements of the ParA dimer upon binding to ATP[31]. We therefore sought to identify if the ParA2_vc structure was altered in the presence of nucleotide. To address this, we performed crystallization trials in the presence of ADP, ATP, and the non-hydrolyzable ATP analogue ATPγS. While crystals grew and diffracted in all co-crystallization experiments, in most cases the crystals possessed the same crystal form as the apo structure described above, and no nucleotide was observed in the active site for these. Nonetheless, one dataset collected on a crystal obtained in the presence of ADP showed a different space group (Table 1). Molecular replacement was performed using the apo ParA2_vc structure, and revealed four molecules per asymmetric unit, consisting of two ParA2_vc dimers (Fig. 2a, Supplementary Fig. 3a).

There is very little difference (RMSD ~0.3 Å) between the two ParA2_vc dimers contained in the ASU (Supplementary Fig. 3b), as well as in the relative subunit orientation between both dimers (Supplementary Fig. 3c). It is also worth mentioning that the quality of the electron density map is significantly better in the ADP-bound structure compared to the apo structure, especially

in the nucleotide-binding site, despite its lower resolution (See Materials and Methods for details). We speculate that this might reflect that the nucleotide stabilizes the ParA2_vc dimer, thus leading to improved diffraction data, as observed previously for P1 ParA[17]. This is also in agreement with thermal melting assay data, which suggests a stabilization of ParA2_vc in the presence of nucleotide[40].

Ligand density was observed in the active site of all four molecules (Fig. 2b), confirming that this structure corresponds to the ADP-bound state of the protein. As shown in Supplementary Fig. 4, the position of the nucleotide is largely similar to that of other ParA orthologues.

As expected, the overall structure of the ParA2_vc monomer in the ADP-bound state is similar to that of apo structure (Fig. 2c), with a RMSD of ~0.8 Å. We do note some changes in the nucleotide-binding site, with the P-loop being better ordered in the ADP-bound conformation. In addition, we observed a change in the positioning of the helix 1, which is closer to that of P1/P7 ParA in the ADP-bound structure. As indicated above, helix 1 forms a domain-swapping interaction with the adjacent molecule in the ParA dimer. Because of the difference in the position of this helix, the architecture of the ParA2_vc dimer differs between the ParA2_vc apo and ADP-states (Fig. 2d), with a slight shift in the relative subunit position between the two states.

**ParA2_vc forms filaments using DNA as a scaffold, regulated by nucleotide hydrolysis.** ParA's ability to form filaments has been highly controversial (see above). ParA filaments have been observed by negative-stain electron microscopy in the presence of nucleotide[21] and/or dsDNA[32–34], but super-resolution fluorescence imaging in cells did not reveal filament formation in

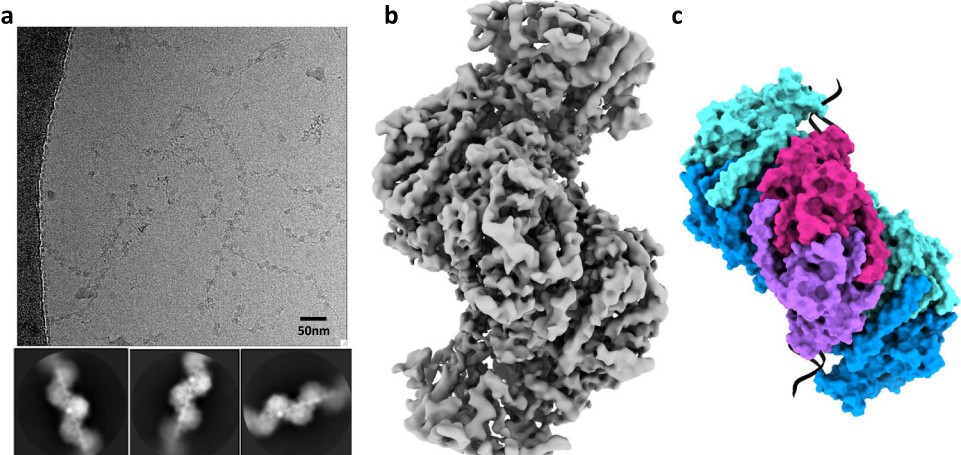

**Fig. 3 Cryo-EM structure of the ParA2$_{vc}$-ATPγS-DNA filament. a** Representative cryo-electron micrograph of the ParA2$_{vc}$-ATPγS-DNA filament (out of a 5785 micrograph dataset), with selected 2D classes underneath. **b** Electron potential map of the ParA2$_{vc}$ATPγS-DNA filament, to 4.5 Å resolution. **c** Atomic model of the filament structure, covering 3 consecutive ParA2$_{vc}$ dimers bound to DNA in cartoon representation, and in the same orientation as the map in **b**. Each ParA2$_{vc}$ molecule is colored separately, with the central dimer in purple and magenta, and the two adjacent dimers in cyan and blue, respectively.

multiple bacterial species, and previously reported crystal structures of ParA-DNA did not provide evidence for higher-order assembly[32,42].

As reported above (Supplementary Fig. 1), we did not observe any ParA2$_{vc}$ filament in the absence of DNA, regardless of nucleotide state. However, in the presence of both DNA and nucleotide, filaments were observed by negative-stain EM (Supplementary Fig. 5a). Intriguingly, we did not observe filaments in the absence of nucleotide, contrary to a previous study[34]. We also observed that the ParA2$_{vc}$-DNA filaments could only be obtained at high protein concentration (protein–DNA ratio ≳5:1 w/w), and that they dissociate at lower protein concentration (protein–DNA ratio ≲5:1 w/w). Furthermore, while the filaments are well ordered and with a clear helical architecture in the presence of ATP, when ADP was used, ParA was visibly bound to DNA, but lacked well-ordered filamentous architecture. In contrast, we were able to obtain stable filaments in the presence of the non-hydrolysable ATP analogue ATPγS, which did not dissociate at lower protein concentration (Supplementary Fig. 5a).

To further investigate the role of ATP hydrolysis for filament formation, we engineered ATP hydrolysis-deficient mutations in ParA2$_{vc}$, as identified previously in the *E. coli* P1 ParA orthologue[43–45]. As shown in Supplementary Fig. 5b, filaments are observed for the ATPase-deficient K124R and K124Q mutants. In contrast, we observed no filaments with the K124E mutant, shown in P1 ParA to lack DNA-binding activity (presumably due to its inability to bind to ATP).

These results suggest that the assembly of the ParA2$_{vc}$-DNA filament is modulated by ATP, with nucleotide binding inducing DNA binding, and filament formation. We postulate that the triphosphate state is required for filament assembly, and ATP hydrolysis triggers its disassembly; however, further biochemical characterization will be required to confirm this. This mechanism is similar to that of Actin and its bacterial homologues, although Actin-like filaments are generally more stable than the ParA2$_{vc}$-DNA filament[46,47].

**Cryo-EM structure of the ParA2$_{vc}$-DNA filament**. We next used cryo-EM to determine the structure of the ParA2$_{vc}$-DNA filaments described above. As indicated, the presence of slowly

hydrolysable ATP analogue was required to obtain stable filaments, suitable for cryo-EM analysis. These filaments readily went into ice (Fig. 3a), and 2D classification of a resulting cryo-EM dataset confirmed that they are ordered, with the DNA backbone, nucleotide, and secondary structure elements of the protein easily identifiable (Fig. 3b). Using this data, we were able to obtain a map to 4.5 Å resolution (Table 2, Fig. 3b, Supplementary Fig. 6), by helical reconstruction. We then exploited the crystal structure of Par2$_{vc}$ bound to ADP (see above) to build an atomic model of the ParA2$_{vc}$-DNA filament (Fig. 3b, Supplementary Fig. 7, and Supplementary movie 1; see materials and methods for details).

As shown in Fig. 3c, ParA2$_{vc}$ forms a left-handed helix, with a rise of 28.68 Å and a twist of −80.57°. This is consistent with the previously reported filament architecture, based on low-resolution negative-stain data[34]. The map includes density for five ParA2$_{vc}$ dimers, and a 48bp-long DNA fragment. Density for the DNA is clearly defined (Supplementary Fig. 7b), with notably some base pair separation in the best-resolved regions of the map. Density for the ATPγS and Mg molecules is also clearly delineated in the active site (Supplementary Fig. 7f).

**ParA2$_{vc}$ interaction with DNA**. Our structure of the ParA2$_{vc}$-DNA filament reveals that each ParA2$_{vc}$ molecule binds to DNA via two interaction sites (Fig. 4a, Supplementary movie 2): (1) In the central ParA domain, three regions (residues 322–328, 345–353, and 376–382, Fig. 4b) interact with the DNA backbone. In particular, a set of basic residues (K326, K327, R350, R352, K376, and K377) form salt bridges with the DNA phosphate. (2) In the N-terminal winged helix-turn-helix domain, the loop between residues 74 and 80 is inserted deep into the minor groove (Fig. 4c). Similarly, several basic residues (K44, K74, H79) form salt bridges with the DNA backbone. Collectively, these basic residues, mostly present at the positively charged end of helices, form a continuous positive surface at the bottom of the ParA2$_{vc}$ dimer, ideally suited for interaction with DNA (Fig. 4d).

Intriguingly, the C-terminal basic residues mentioned above are not conserved across ParA orthologues, and the N-terminal residues also lack conservation within type Ia orthologues, as shown in Supplementary Fig. 8. Nonetheless, for two of the corresponding regions (residues 322–328 and 345–353), basic

**Table 2 Cryo-EM data collection, refinement, and validation statistics.**

| | ParA2$_{vc}$-ATPγS-DNA (EMDB-12515) (PDB 7NPF) |
|---|---|
| *Data collection and processing* | |
| Magnification | 130,000 |
| Voltage (kV) | 300 |
| Electron exposure (e–/Å²) | 52.02 |
| Defocus range (μm) | −2.3-1.3 |
| Pixel size (Å) | 1.047 |
| Symmetry imposed | Helical |
| Initial particle images (no.) | 390,604 |
| Final particle images (no.) | 182,997 |
| Map resolution (Å) | 4.5 |
| FSC threshold | 0.143 |
| Map resolution range (Å) | 3.3-5.8 |
| *Refinement* | |
| Initial model used (PDB code) | 7NPE |
| Map sharpening B factor (Å²) | 40 |
| Model composition | |
| Non-hydrogen atoms | 3330 |
| Protein residues | 3216 |
| Ligands | 16 |
| Nucleic acid | 98 |
| *B factors (Å²)* | |
| Protein | 111.36 |
| Ligand | 90.37 |
| Nucleic acid | 137.96 |
| *R.m.s. deviations* | |
| Bond lengths (Å) | 0.004 |
| Bond angles (°) | 0.917 |
| Validation | |
| MolProbity score | 2.28 |
| Clashscore | 18.28 |
| Poor rotamers (%) | 0.47 |
| Ramachandran plot | |
| Favored (%) | 91.1 |
| Allowed (%) | 8.77 |
| Disallowed (%) | 0.13 |

residues are present in all ParA sequences, suggesting that the mode of binding is conserved. Furthermore, co-evolution analysis shows that several of the residues involved in the interaction with DNA, notably K327, R350, R352, K377, have significant evolutionary links with other residues at or near the DNA-binding region (Supplementary Data 1). In keeping with this, the recently published structure of a type Ib ParA protein (the *Helicobacter pylori* Soj protein, *Hp*Soj) bound to DNA[32], revealed a largely similar set of interactions with the nucleic acid backbone (Fig. 4e). In contrast, the N-terminal domain is not present in type Ib ParA proteins, and accordingly this set of interactions is not present in the *Hp*Soj-DNA structure. Similarly, while a number of basic residues are found in region 376–382 of type Ia ParA proteins, this region (a loop and part of helix 16) is not found in type Ib ParA proteins, with the exception of the *C. crescentus* ParA orthologue (Supplementary Fig. 8). As shown in Fig. 4b, e, this loop forms a deep insertion within the major groove of the DNA, causing significant distortion of its backbone. As a consequence, the relative orientation of the DNA molecule differs significantly between the *Hp*Soj-DNA crystal structure[32] and the ParA2$_{vc}$-DNA structure reported here (Supplementary Fig. 9). Based on the sequence alignment, this difference in DNA orientation can likely be generalized between type Ia and type Ib ParA orthologues, and might be related to the transcription repression activity of type Ia ParA proteins[18]. As mentioned above, the *C. crescentus* ParA orthologue (which belongs to the

type Ib family) possesses the additional DNA-binding region near the C-terminus normally found only in type Ia orthologues, and may therefore share some common properties between the two families.

It should also be mentioned that the crystal structure of the archaeal plasmid pNOB8 ParA protein[48] revealed a completely different binding mode to both *Hp*Soj and ParA2$_{vc}$ (Supplementary Fig. 9). In spite of this, the sequence alignment shown in Supplementary Fig. 8 indicates that while this protein does not possess the N-terminal domain of type Ia ParA proteins, all three DNA-binding regions of the core domain are present in the pNOB8 ParA protein, and include several basic residues. It is not known if the difference in DNA interaction corresponds to a crystallization artefact, or reflects biological differences in the interaction with DNA between archaeal and bacterial ParA proteins.

**Filament assembly interface, and remodelling of the ParA dimer upon filament assembly.** In the structure of the ParA2$_{vc}$-DNA filament reported here, adjacent ParA dimers form extensive contacts (Fig. 5a), with a surface area of ~1500 Å². This interface is largely mediated by three regions: two helices, located at the C-terminus (residues 325–339 and 381–405), and a helix-turn-helix motif from the N-terminal domain (Fig. 5b, Supplementary movie 2), forming electrostatic contacts (Supplementary Fig. 10a). In particular, helices 14 and 16 possess a number of exposed charged residues at the oligomerization interface (Supplementary Fig. 10b), that form salt bridges with the adjacent subunit. We note that it had previously been proposed that only type Ia ParA orthologues could form filaments, which would be formed only by interactions via the N-terminal domain[34]. However, our structure does not support this, and most of the filament oligomeric interface is located in the C-terminal region of the protein (Fig. 5b, Supplementary Fig. 10).

Intriguingly, the residues involved in the interface between ParA2$_{vc}$ dimers, within the filament, are not conserved across orthologues, even within the type Ia family (Supplementary Fig. 8). This could indicate that the filament architecture differs in other bacteria, and/or that some ParA orthologues may not form filaments. Nonetheless, charged residues are found in similar positions in most sequences, and many of these have significant co-evolution links (Supplementary Data 1). In order to verify the role of these residues in the filamentous architecture of ParA, we engineered a point mutation (K388A) onto one of these residues. As shown in Supplementary Fig. 11, the resulting protein is not able to form rigid filaments in the presence of ATP and DNA, confirming that this residue is critical to the dimer–dimer interface.

Collectively, this evidence supports the hypothesis that the oligomerization interface is conserved across ParA proteins encoded by chomosomes and plasmids. Nonetheless, further structural studies of filament architectures in other ParA proteins would be required to verify this.

Finally, another striking feature in the ParA2$_{vc}$-DNA filament structure, is the difference in conformation of ParA molecules, compared to the crystal structures described above. Specifically, helix 1 undergoes a striking conformational change, merging with helix 2 to form a single, extended helix ~15 Å from its position in the structures obtained without DNA (Fig. 5c). As indicated above, helix 1 forms a cross-dimer interaction, in the ParA2$_{vc}$ dimer. As a consequence, the angle between the two molecules in the filament structure is altered by ~30 degrees, compared to the crystal structures described above (Fig. 5d, Supplementary movie 3).

This structural rearrangement likely explains the cooperativity in DNA binding, observed in many ParA orthologues (see

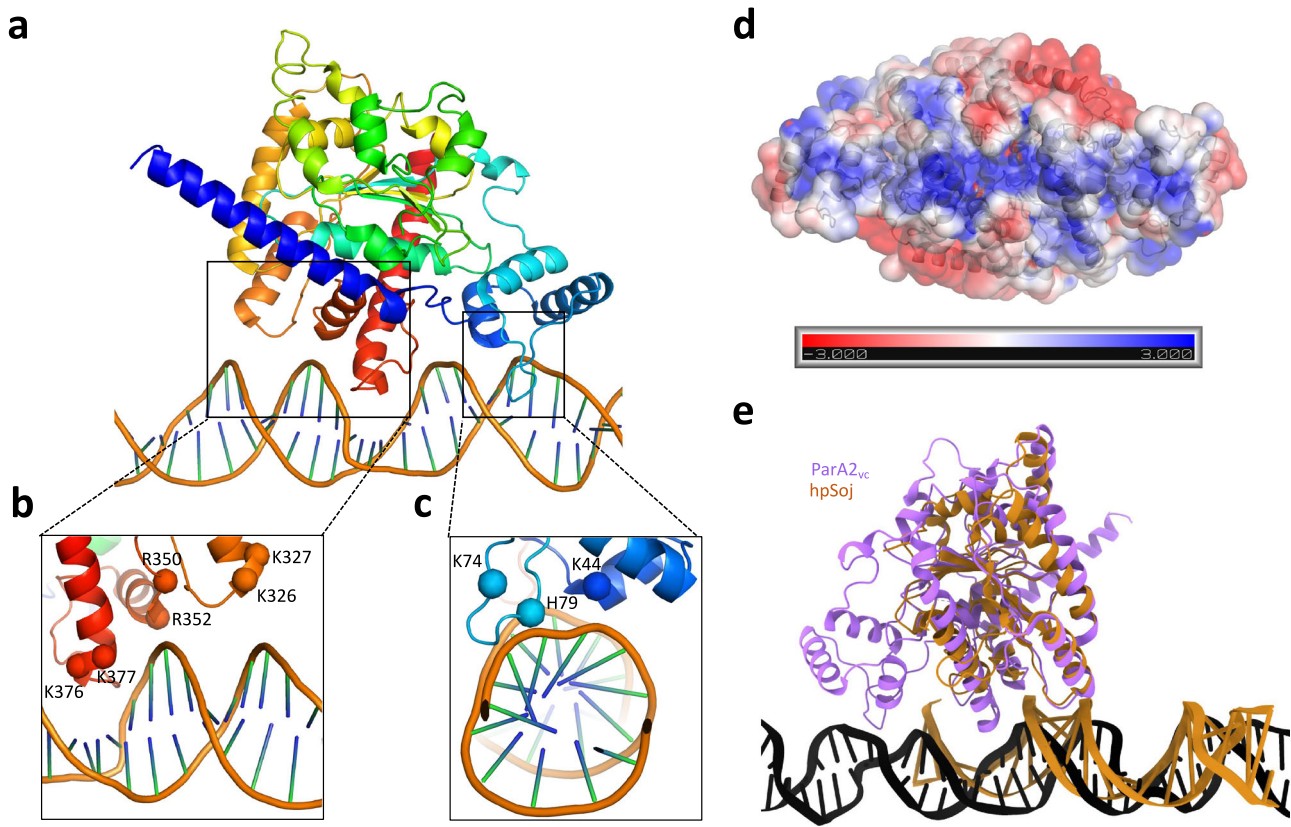

**Fig. 4 Structural basis for ParA2_vc's interaction with DNA. a** A ParA2_vc monomer, and the DNA molecule, from the cryo-EM structure are shown in cartoon representation, in rainbow coloring. The two regions forming contacts with the DNA are in black boxes. Close-up views of these two regions are shown in **b** and **c**, with the basic residues forming salt bridges with the DNA backbone indicated. **d** Electrostatic surface representation of the ParA2_vc dimer. A positively charged stretch is clearly present, corresponding to the DNA-binding surface. **e** A ParA2_vc monomer bound to DNA is shown in purple, overlaid to a *hp*Soj protein bound to DNA in orange. Both proteins interact with DNA on the same surface, but ParA2_vc forms additional contacts, via the NTD and the C-terminal helix.

discussion), and which have been proposed to be critical for chromosome segregation[49]. Specifically, we propose that the remodeling of the ParA dimer upon DNA binding increases its binding affinity, as an additional binding surface is formed for the binding of additional ParA dimers. We note that residues in helix 1 are not conserved (Supplementary Fig. 8), as observed for the DNA binding and filament interface (see above), but these residues show strong co-evolution links to other residues, closely positioned in the adjacent molecules within the filament structure (Supplementary Data 1), which suggests that the remodelling of helix 1 upon DNA binding is likely applicable to other type Ia ParA orthologues.

To further investigate the role of helix 1 in filament formation, we engineered a deletion mutant of ParA2_vc where the entire helix 1 was removed (Δ3-36). Intriguingly, we observed that the resulting protein maintained its ability to form filaments in the presence of nucleotide and DNA. This demonstrates that the N-terminal helix is not essential for filament formation, and consistent with the hypothesis that both type Ia and type Ib ParAs (the later of which lack the NTD, including helix 1) are able to form filaments[33].

## Discussion

In this study, we have reported the structure of the ParA2_vc protein, in three states: apo, nucleotide-bound (ADP) and in a filamentous complex with nucleotide and DNA. Importantly, we report the first structure of a ParA protein in the filamentous form. This structure allows us to identify how ParA molecules interact with the DNA, but also how they form higher-order structures. In particular, we show that the NTD forms additional contacts with the DNA, revealing differences between type Ia and type Ib ParA proteins. In contrast, we show that the higher-order oligomerization is mostly mediated by the C-terminal region, and co-evolution data suggests that this interface is conserved across ParA orthologues.

From the three structures reported here, we are able to observe the conformational change occurring upon nucleotide binding and filament formation. Combined with prior biochemical and cell-based assays reported previously[34,40], these structures allow us to propose a mechanistic model for ParA's higher-order assembly, shown in Fig. 6: (a) At physiological concentrations, ParA is at equilibrium between monomeric and dimeric state in the absence of nucleotide. The recruitment of ATP stabilises the dimer. (b) A nucleotide-bound ParA dimer can bind to DNA, and this interaction induces a conformational change to the dimer architecture. (c) This change exposes the filament-forming surface of the DNA-bound ParA dimer, leading to the formation of a filament along the DNA. When encountering a *parS*-bound ParB, this activates ParA's ATPase activity, leading to disassembly from the DNA, coupled with the release of hydrolysed nucleotide (Fig. 6).

We note that previous biochemical data have shown that ParA2_vc binds cooperatively to DNA[40], as also observed in other ParA orthologues[18,23,33]. The structures reported here likely provide a mechanism for this cooperativity, with the structural changes associated with DNA binding allowing to form a charged

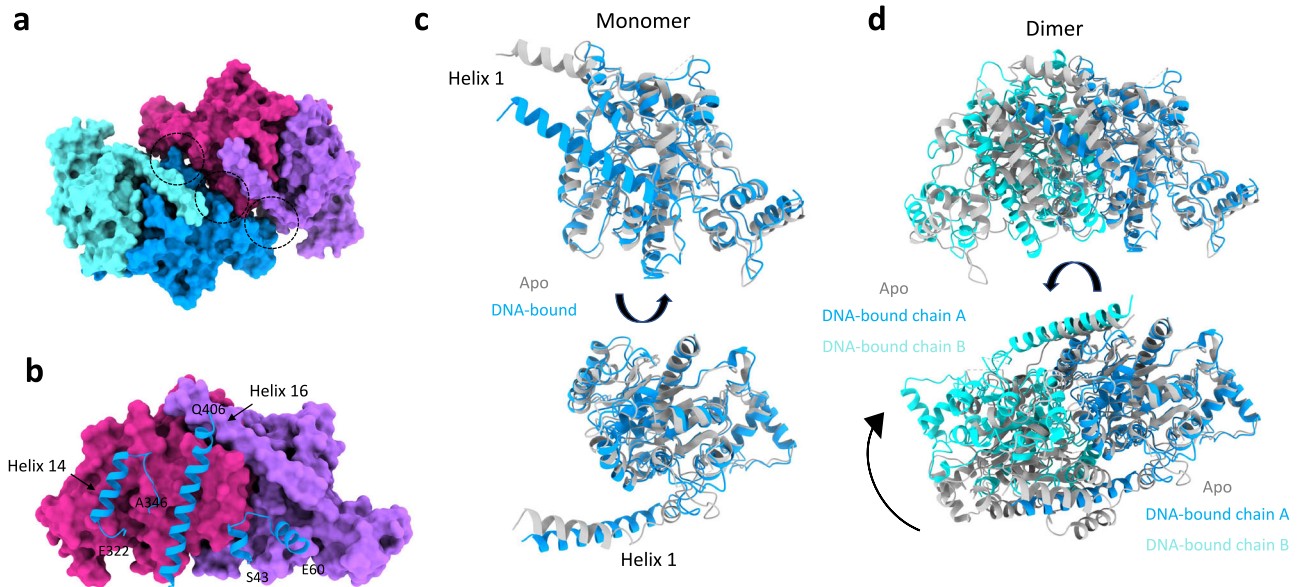

**Fig. 5 ParA2$_{vc}$ filament interfaces, and structural changes upon filamentation. a** Surface representation of two adjacent ParA2$_{vc}$ dimers in the filament structure, colored as in Fig. 3c. The interacting regions are indicated with black dotted circles. **b** One dimer from **a** is shown in surface representation, and the regions of the second dimer that are involved in the interaction are shown in cartoon. The residue boundaries are indicated. **c, d** Comparison between the ParA2$_{vc}$ structure in the free (grey) and DNA-bound (blue/cyan) conformations. A monomer is shown in **c**, illustrating the rearrangement of helix 1; and a dimer is shown in **d**, aligned on the blue subunit, to show the dramatic change in the dimer architecture.

surface that permits electrostatic interactions with adjacent dimers. This leads to an increased affinity for the binding of additional ParA2$_{vc}$ molecules adjacent to it. It remains to be verified if the change in dimer architecture is a result of the binding to ATP$_\gamma$S, or to DNA.

As mentioned above, whether ParA proteins do form filaments, and the role of such filaments, has remained controversial. In particular, multiple studies using fluorescently tagged ParA orthologues in dividing cells, revealed that it clusters at high-density chromosomal regions (HDRs)[18,40,50,51], and do not form filaments across the cell, as required for a mitotic-like mechanism. In keeping with this, the negative-stain EM experiments reported here suggests that at near-physiological concentration, the ParA2$_{vc}$ filaments can only form when bound to non-hydrolysed ATP. We therefore propose that in situ, ParA proteins merely form small patches of filaments along the DNA, corresponding to those HDRs observed by super-resolution fluorescence microscopy. This likely helps forming high-density ParA regions in the nucleoid, as required in the proposed diffusion-ratchet model for segregation.

Nonetheless, a number of questions remain to be addressed to fully validate this mechanistic model. Specifically, while we observed major changes in the dimer architecture from the free ParA to the filament state, it remains to be established if these changes are induced by DNA binding, or by the recruitment of adjacent ParA molecules during filament formation. Furthermore, as indicated above, sequence similarity between ParA orthologue is low, and the residues at the DNA-binding regions and filament interface are mostly not conserved. While co-evolution analysis (Supplementary Dataset 1) and biochemical studies suggests that these features are likely maintained across species, this remains to be verified experimentally.

It is also worth noting that ParA is structurally similar to MinD, also a P-loop walker type ATPase, but involved in Z-ring localization[52]. MinD forms a dimer, structurally similar to ParA, but does not bind to DNA. Nonetheless, it was recently shown that MinD forms filaments, in the presence of its interacting

partner MinC[53]. However, comparison of the MinCD filament to our cryo-EM structure of the ParA2$_{vc}$-ATP$_\gamma$S-DNA filament (Supplementary Fig. 12) reveals that the filaments formed by these two proteins have a completely different architecture, and use different interfaces to form dimer–dimer contacts. Based on this, we postulate that filament formation is not a general feature of this family of proteins, but was adopted independently in ParA and MinD proteins, during evolution.

## Methods

**Protein expression and purification.** For ParA2$_{vc}$ (and mutants) purification, we used the procedure described in Chodha et al.[40], with a number of modifications. Briefly, cells containing a plasmid including the *parA2$_{vc}$* gene with a T7 promoter, were grown to log phase, expression was induced by adding 1 mM IPTG, the protein was expressed at 16 °C overnight, and cells were centrifuged at 6000 *g* for 15 min. Cell pellets were resuspended in sonication buffer (50 mM Tris-HCL pH8, 100 mM NaCl, 0.1 mM EDTA, 2 mM DTT, 50 mM (NH$_4$)$_2$SO$_4$) supplemented with 1 mg/ml of lysozyme and ½ Roche protease inhibitor tablet (10 ml/g of cell pellet). The obtained sample was lysed via sonication for 10 min (in 30 s intervals). For electron microscopy experiments, lysed cells were centrifuged at 26,000 x *g* for 25 min, 0.35 g/ml of (NH$_4$)$_2$SO$_4$ was added to the supernatant, which was centrifuged as above. The pellet was resuspended in 20 ml of Buffer A (50 mM Tris pH8, 100 mM NaCl, 0.1 mM EDTA, 2 mM DTT and 20% glycerol), and dialysed against 2 l of buffer A using 10,000 MWCO SnakeSkin® Dialysis tubing (Thermo Fisher) overnight at 4 °C. The sample was then ran through a Heparin HiTrap column (Sigma), and eluted in Buffer A supplemented with 1 M NaCl. Fractions containing ParA2$_{vc}$ were then run through a MonoQ column in buffer A and eluted in Buffer A supplemented with 1 M NaCl. Finally, the sample was run through a 16/600 pg 200 superdex column (Sigma), in storage buffer (50 mM Tris pH8, 500 mM NaCl, 0.1 mM EDTA, 2 mM DTT). Fractions containing ParA2$_{vc}$ were concentrated using a VivaSpin 20 column 10,000 MWCO (Sartorius), before snap freezing in liquid nitrogen for storage at −80 °C. For crystallography, pellets were treated following the method outlined by Chodha et al.[40]. ParA2vc mutants were engineered in the expression plasmid by site-directed mutagenesis using the QuickChangeII kit (Agilent).

**Crystallization, X-ray crystallography, and structure refinement.** ParA2$_{vc}$ at 20 mg/ml was set up for crystallisation trials in sitting drop 96-well plates (Hampton Research), in standard crystallization screens (QIAgen) using a mosquito protein crystallisation robot (sptlabtech), and incubated at 4 °C and 20 °C for each screen. Crystals were obtained in many conditions; however, a significant number of them had a needle-like morphology, and did not diffract. Nonetheless,

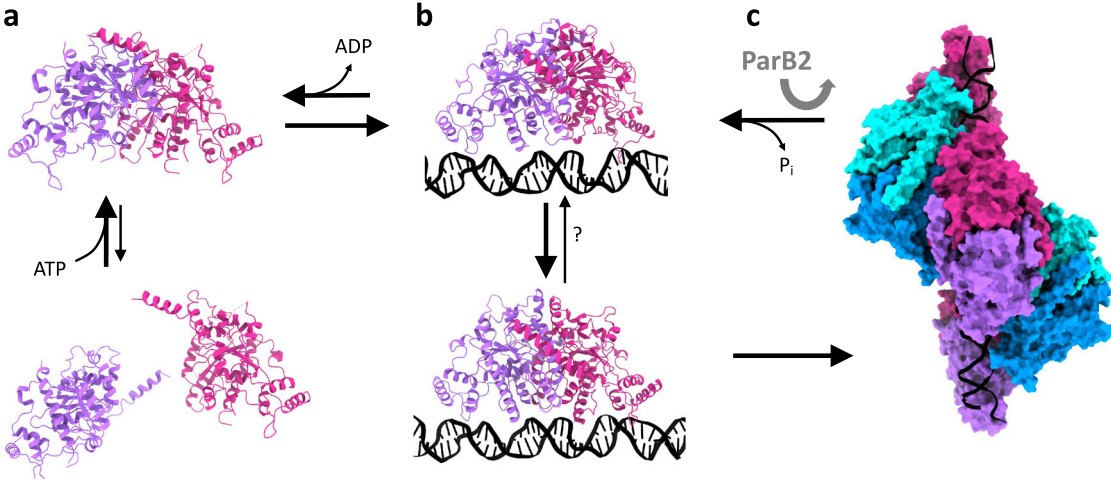

**Fig. 6 Proposed mechanism for ParA's cooperative binding to DNA, regulated by ParB. a** In isolation, ParA is in equilibrium between monomer and dimer, with the dimer stabilized by the recruitment of nucleotide. **b** In the presence of DNA, the dimer undergoes a dramatic architecture change, exposing its oligomerization interface. **c** This leads to the formation of a higher-order assembly, in the form of a short filament segment. The presence of ParB-bound cargo stimulates ParA's ATPase activity, which leads to its return to the DNA-free conformation of the dimer. This in turn leads to the dissociation of ParA from the DNA, and the release of ADP.

we were able to obtain crystals in 0.1 M tri-sodium citrate pH 5.5, 20% w/v PEG 3000, grown at 20 °C, with a different morphology, which diffracted to ~2.5 Å. A dataset was collected for these under cryo conditions, at the Diamond Light Source beamline io3, at a wavelength of 0.976 Å. The diffraction data was processed to 2.6 Å, and was indexed to the space group P3₂ 1 2. Phases were obtained by molecular replacement with Phaser[54] implemented in the Phenix package, using a homology model of ParA2_vc based on the P7 ParA structure (3ezF), and missing of the N-terminal residues 1–107. The N-terminal domain was then manually built into the electron density map using *Coot*[55]. The obtained atomic model was refined through multiple cycles of building and refinement, using phenix.refine[56] and Isolde[57], to final Rwork/Rfree values of 27.5%/33.5%, respectively (see Table 1). The coordinates have been deposited to the PDB, under the accession number 7NPD. We acknowledge that these refinement statistics are relatively poor, and that the geometry of the obtained model is not ideal for the reported resolution, in spite of many cycles of refinement and re-building. Notably we have only ~83% of residues in the favored region of the Ramachandran plot, with 3.7% of outliers (13% allowed). This is likely because of the poor quality of the low-resolution data, leading to a poor-quality map and low restraints for refinement. We emphasize that we collected multiple datasets from similar crystals, which all showed similar defects. We therefore suspect that the low quality of the data is due to the flexibility of the molecule, in the absence of ligands in the active site.

For the nucleotide-bound structure, crystallization trials were set up as above, with the ParA2vc sample co-crystallized with 5 mM ADP, ATP or ATPγs and 5 mM MgCl₂. As above, crystals were observed in many conditions; however, they mostly belong to the needle-like morphology for which no diffraction was obtained, or to the same space group as above. Several datasets were collected nonetheless, but none showed density for ligands in the active site. However, a different crystal morphology was obtained in 24% w/v PEG 1500 with 20% w/v glycerol at 4 °C in the presence of ADP. A dataset for such crystal was collected under cryo conditions at the Diamond Light Source beamline io3 at a wavelength of 0.976 Å, and diffracted to ~3 Å. The dataset was processed to 3.2 Å, and revealed a space group of P61 2 2. Molecular replacement was carried out using the structure obtained from the data set, as described above, and the structure was refined as described above, to final Rwork/Rfree values of 23.8%/27.8%, respectively, with 91.5% residues in the favourable region of the Ramachandran plot, 6.8% in the allowed region, and 1.6% outliers (Table 1). We note the relatively high average B-factor for this structure, largely due to the fact that the N-terminal domain is less well resolved, and as a consequence the B-factor for this domain is around 160. The coordinates have been deposited to the PDB, under the accession number 7NPE.

**Negative-stain EM.** For the ParA2_vc samples, 1 mg/ml of protein was incubated with 3.5 mM nucleotide and MgCl₂ at 30 °C for 15 min. 1/100 dilutions were made for each nucleotide state into sample buffer containing 100 mM NaCl, 50 mM Tris pH7.5, 2.5 mM nucleotide, and 2.5 μM MgCl₂ before staining in 0.75% uranyl formate on glow discharged carbon-coated copper grids (Agar scientific).

For the ParA2_vc-ATPγS-DNA filaments, the samples were prepared as above, except with 2 mg/ml of ParA2_vc. Each sample was then spiked with sonicated salmon sperm DNA (sssNA) (length 1 kb) to a final concentration of 0.2 mg/ml and left at 30 °C for a further 10 min, leaving a ParA2_vc:DNA ratio of ~6:1 and nucleotide and MgCl₂ at 3 mM. A 1/10 dilution for each state was made using

sample buffer and staining was done as above. For the ATPγS nucleotide state, ParA2_vc at 1 mg/ml was incubated with MgCl₂ and ATPγS at 4.5 mM and incubated at 30 °C for 15 min. The sample was then spiked with sssDNA (1 kb) to a final concentration of 0.14 mg/ml and incubated again, resulting in a ParA:DNA ratio of ~3.5:1. From this stock a 1/10 dilution was made, and grid preparation was done as above.

All negative-stain data was collected manually at a defocus range of ~−1 μm to −3.5 μm on a CM100 TEM (Phillips) with a MSC 794 camera (Gatan) at 27,500x with a pixel size of 7.2 Å. For 2D classification, the micrographs were processed using cisTEM[58], and 2D classes were generated using a box size of 24 pixels.

**Cryo-EM data collection, processing, and model building.** ParA2_vc at 4 mg/ml was incubated with ~ 6 mM ATPγS and MgCl₂ at 30 °C for 15 min, and was then spiked with sssDNA to a final concentration of ~0.8 mg/ml of DNA, ~1.9 mg/ml of ParA, and ~4 mM of ATPγS and MgCl₂. The sample then furtherly incubated for 10 min at 30 °C. Tween 20 was then added to a final concentration of 0.1% before grid preparation, to facilitate incorporation of the filaments into the holes of the carbon grid. Three microlitres of this sample was applied to glow discharged 300 mesh Quantifoil R2/2 grid and left for 30 s before manually blotting with filter paper. The grid was then loaded into a Leica EM-GP plunge freezer at 80% humidity and 4 °C where a further 3 μl was applied for a 10 s incubation followed by a 4 s blot before being plunged into liquid ethane. Micrographs of ParA2_vc-ATPγS-DNA filaments were recorded using EPU (Thermo Fischer), on a 300KV Titan Krios FEG microscope with a Gatan K2 Summit detector in counting mode. 5785 movies were recorded with a pixel size of 1.047 Å over 50 frames with a total dose of 52.02 e⁻/Å², at a defocus range of −2.3 μm to −1.3 μm. Frames were aligned using MotionCor2[59], and CTF estimation was obtained using CTFFIND4.1[60]. Filaments were then manually picked in Relion[61], and helical segments were extracted with a box size of 200 pixels, a tube diameter of 140 Å, and a helical rise of 30 Å. 2D classification, 3D classification, and 3D refinement was performed in Relion-3[62], with a tube density used as an initial reference map. For 3D classification, 6 classes were generated over 25 iterations with a mask diameter of 200 Å over a local helical search range of −70° to −90° for twist and 20 Å to 40 Å for rise. A chosen class was then selected and put into 3D refinement following the same parameters, and the resulting map was post-processed in Phenix[63] (see Supplementary Fig. 6 for the detailed pipeline). The local resolution was determined with ResMap[64].

To build the atomic model, a polyA-polyT dsDNA molecule was generated in Coot, and placed in the map. A ParA2_vc dimer from the ADP-bound structure (see above) was fitted into the EM map using ChimeraX[65], and helix 1 was re-built manually in Coot. Multiple copies of the resulting dimer were then generated, and placed in the corresponding density in ChimeraX. The final model was then subjected to real-space refinement in Phenix[66]. The coordinates have been deposited to the PDB, under the accession number 7NPF, and the map has been deposited to the EMDB, with the accession number 12515.

**Sequence alignment, co-evolution, and structure representation.** The ParA orthologue sequences were aligned with ClustalW[67], and ESPript[68] was used to generate the alignment figure. The co-evolution analysis was performed using the

GREMLIN server[69]. All structural figures and movies were generated using either ChimeraX, or PyMOL[70].

**Reporting summary**. Further information on research design is available in the Nature Research Reporting Summary linked to this article.

## Data availability statement

The map for the ParA-ATPgS-DNA cryo-EM structure was deposited in the EMDB, accession number EMD-12515. The corresponding atomic coordinates were deposited on the PDB, accession number 7NPF. The structure factors and atomic coordinates for the ParA2 apo and ADP-bound crystal structures were deposited in the PDB, with accession numbers 7NPD and 7NPE, respectively. The molecular replacement for the ParA2 structure was performed using the P1 ParA crystal structure as a search model, accession number 3EZ6. All other data generated in this study are provided in the Supplementary Information and Supplementary Data files. Supplementary Movies 1–3 are available with the paper online.

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

## Acknowledgements

A.V.P. was recipient of a PhD scholarship from the Global Strategic Alliance at the University of Sheffield. We are grateful to Dr Satpal Chodha for helpful discussion on ParA biochemistry D.M. was supported by BBSRC grant BB/R019061/1 (to J.R.C.B.). We acknowledge the University of Sheffield EM facility for assistance with negative-stain EM data collection, and cryo-EM grid screening. X-ray crystallography data for the ParA2$_{vc}$ apo and ADP-bound were collected at the Diamond Light Source (proposal MX24447), and the Cryo-EM data for the ParA2$_{vc}$-DNA structure was collected at eBIC (proposal EM20970).

## Author contributions

A.V.P., L.C.H, and J.R.C.B. conceived the project and designed the experiments. A.V.P. performed the protein expression and purification, crystallization, X-ray crystallography, negative-stain EM, and cryo-EM experiments, processed the crystallography and EM data, and performed the atomic model building and refinements. S.B.T. aided with the EM screening and optimization, and DM contributed to the cryo-EM data refinement. L.C.H. and A.V.P. constructed plasmids. A.V.P. and J.R.C.B. wrote the manuscript, with contributions from all the authors.

## Competing interests

The authors declare no competing interests.
