## [Peer Review File · Nature Communications]

The structure of the bacterial DNA segregation ATPase filament reveals the conformational plasticity of ParA upon DNA binding.TitleREVIEWER COMMENTS

Reviewer #1 (Remarks to the Author):

This manuscript by Parker and colleagues presents an interesting structural characterisation of the ParA2 segregation protein encoded by *Vibrio cholerae* chromosome 2. The authors describe the crystal structure of ParA2 in the apo and ADP-bound state and the cryo-EM structure of ParA2 filaments assembled on non-specific DNA.

A number of ParA protein structures have been characterised and described in the literature. The solution of ParA2VC allows some interesting comparison to be drawn and contributes to a better understanding of this family of Walker-type ATPases. The element of novelty of this work comes from the determination of the ParA2 filament structure that has not been previously described for any ParA protein. The details revealed by this structure shed new light on ParA dimer-dimer and ParA-DNA interactions. Overall, these findings contribute a new perspective to the field of genome segregation.

The storyline is clear and logical. Experiments appear to have been rigorously performed and include appropriate controls. The results support the conclusions. Figures are well organised and annotated. The videos are interesting and help to a large extent to visualize the three-dimensional details of the structures.

I have some queries about some experiments and minor suggestions to improve the manuscript.

1. Figure S5, EM of ParA2 in the presence of DNA and nucleotides: the filaments observed in the presence of ATP or ADP or ATP-gamma-S appear quite different. The ParA2-ATP filaments appear dark positively stained, whereas the filamentous structures visible in the ParA2-ADP panel are negatively stained and the ParA2-ATP-gamma-S are again quite different. Can the authors clarify and rationalize these differences?
2. The authors state that ParA2-DNA filaments could only be obtained at high protein concentration, as they dissociate at lower protein concentration. I am curious as to why ParA2-ADP-DNA filaments would be unstable, as in this scenario there is no hydrolysis. I guess there are other factors affecting the stability.
3. Have the authors tried to perform these EM experiments using a ParA2 mutant impaired in ATP binding? If the DNA binding requires ATP binding as a prerequisite, then one would expect to observe no filaments. Additionally, the observation of an ATPase defective ParA2 mutant would provide support to the hypothesis that ATP hydrolysis triggers the ParA2-DNA filament disassembly.
4. Abstract: line 20, 'parS' should be italicized.
5. Introduction: line 37, 'The Par loci' should be 'The par loci'.
6. For consistency, use always upper or lower case when describing type I or type II and III systems throughout the manuscript.
7. Introduction: line 44, the system does not encode the centromere site; I would suggest to change the sentence to 'The type I segregation system locus encodes the ATPase ParA; an adapter protein, ParB; and contains centromere-like parS site(s)'.
8. Introduction: line 52, change to 'ParA forms...'
9. Introduction: line 57, delete 'in'
10. Introduction: line 82, upper case for Gram-negative.
11. Introduction: line 85, delete 'having'
12. Introduction: line 92, no need for upper case for 'Chromosome 2'.
13. Results: line 114, rephrase the sentence, P1 should not be Italic.
14. Results: line 116, I would suggest to change the sentence to 'to characterize ParA2vc structure, to verify if it adopts a similar architecture...'
15. Results: line 153, it should be P7 rather than T7.
16. Results: line 158, no need of period at the end of the heading here and elsewhere.
17. Results: line 251, Italic for 'Helicobacter pylori'.

18. Results: line 265, upper case for 'c-terminus'.
19. Results: line 295, the sentence can be better worded, for example 'Collectively, this evidence supports the hypothesis that the oligomerization interface is conserved across ParA proteins encoded by chromosomes and plasmids'.
20. Discussion: lines 334-5, 'When encountering a DNA cargo bound to DNA'...this sentence is not clear, it needs clarification.
21. Discussion: line 335, 'leading to disassembly from the DNA'.
22. Discussion: from line 367 to 376, a different referencing style is adopted and two references are cited using author names. This needs to be rectified.

Reviewer #2 (Remarks to the Author):

This manuscript "The cryo-EM structure of the bacterial 1 type I DNA segregation ATPase filament reveals its conformational plasticity upon DNA binding.", by Parker et al., investigates the structural properties of the partitioning protein ParA from the chromosome 2 of *Vibrio cholera* (ParA2). It reports the crystal structure of ParA2 in several nucleotide bound states, and the structure of ParA2-ATPgS bound to DNA by cryo-EM at high resolution (4.5 angstrom). They build an atomic model of ParA2-DNA filament using the ParA2-ADP crystal structure. It reveals large conformational changes in the ATPgS and DNA bound form allowing the authors to provide a mechanistic model for the cooperative DNA binding of ParA.

The manuscript is well written and data are clearly presented. The data presented are largely confirmatory to what is already known for type Ia ParA in general and ParA2 in particular. The reported structures are highly similar to those of ParA from plasmids P1/P7 to which ParA2 is a close homolog. The filament formation on DNA, in a left-handed helical conformation, was also described earlier. However, authors applied cryo-EM technique, which provide with a very resolution allowing solving the structure of ParA2 in the filament form. It provides the numerous ParA-DNA contacts. They also detect a large conformational change involving the helix 1 and 2. They proposed that this conformational rearrangement explain the cooperativity of ParA on non-specific DNA, and based on amino-acids conservation that this property may be share amongst other type Ia ParA. This structural changes in the nucleotide bound form associated with DNA binding allow the formation of a charged surface that permits electrostatic interactions for adjacent dimers. The authors did not know whether this rearrangement is due to ATPgS binding, DNA binding or both.

Lastly, they discussed that the filaments formed by ParA2 and MinD (a member of the ParA/MinD superfamily) display completely different architecture and used different interfaces for dimer-dimer contacts, thus that the filament formation by these two distinct subfamilies are evolutionary unrelated. This study is well performed and report the structure of ParA2 in its "filament" conformation. It also provided an explanation, based on a large conformation change, for the DNA binding cooperativity. While this is very interesting, most of the data are confirmatory and best suited in a more specialized journal.

The Table S1 is provided without title and legend.

Minor comments:

Introduction

§1: "motor protein" is not defined; are these true motor proteins as in Euk or motor-like based on

activity or mode of action; "motor" has to be defined.

§2: ParB also display sequence independent DNA binding activity, as shown for P1/P7 but also for chromosomal ParB. This should be corrected.

§2: "ParB stimulates Par's ATPase activity". Some mechanistic details have been shown for this stimulation through arginine-finger like motif. At least, references for this more recent mechanistic detail should be included.

§5: "in type Ib systems, parS is located after the promoter, followed by parA and parB." Is this true for all Ib type? Some reference would be welcome.

§7: The size/unit of Chr1 and Chr2 has to be corrected.

Throughout the manuscript, for stating that no filament formation has been revealed in vivo, it would gain in precision to mention and reference explicitly that experiments has been performed by super resolution microscopy instead of "fluorescence microscopy" or "fluorescence imaging" since these latter does not have the resolution capability for this argument.

Results

p. 8, last §: "they dissociate at lower protein concentration (Figure S5): This is not shown directly in the Figure. At least, the figure legend should be completed by indicating what experimental conditions have been tested and describe the results.

p. 11: "the basic residues mentioned above are largely not conserved across ParA orthologues". These basic residues (K44, K74, H79) are in the Nterm domain, which is not present in type Ib ParAs. They could therefore not be considered conserved or not. This sentence should be clarified.

Reviewer #3 (Remarks to the Author):

This manuscript presents X-ray and cryo-EM studies of *Vibrio cholerae* ParA. The crystal structures reported captured various nucleotide states of ParA dimer, while cryo-EM of the ParA filament, in the presence of DNA, is the first near-atomic resolution structure of the ParA filament. An earlier structure for the ParAvc filament was reported by Hui et al in 2010 (PNAS), but that was done at very low resolution in negative stain. The authors are able to show that the ParA dimer is stabilized by nucleotide binding and observe a dramatic structural rearrangement upon DNA binding and filament assembly. Below are some comments and suggestions that need to be addressed in a revised manuscript.

Major point:

(1) My biggest concern is the model quality for both X-ray models and cryo-EM models. This can be easily judged from the statistics table. For crystal structures, extremely low Ramachandran favored (%), 83% and 89%, are reported. These are even worse than most recent cryo-EM models. Both models also have unreasonably high atom B-factors (85 and 124). Finally, the high clashscore (44 and 15) and high Bond angle/length RMSD also confirm the models were poorly refined.

(2) There are similar issues in the cryo-EM model: terrible Ramachandran, Rotamer, clashscore statistics all indicate the model has been poorly refined. It is the authors' responsibility to properly refine a model.

(3) Only a map:map FSC is provided for the cryo-EM resolution estimation. This is often useless, as people have shown in many cases that this only measures the consistency between two half maps and can give a 4.0 Å resolution estimation even when the helical symmetry is wrong. A Model:map FSC must also be provided for resolution estimation.

Minor points

(4) A monomer docked into the 4.5Å cryo-EM map needs to be shown in the paper. The map should

be adjusted to higher threshold than what is shown in Figure S7, so the readers can tell the quality of the map better.

(5) ATP- γ -S is referred to as a non-hydrolyzable analog. This is not true, and it is a slowly hydrolyzable analog.

(6) line 352: "In keep with this" should be "In keeping with this"

Reviewer #1 (Remarks to the Author):

This manuscript by Parker and colleagues presents an interesting structural characterisation of the ParA2 segregation protein encoded by *Vibrio cholerae* chromosome 2. The authors describe the crystal structure of ParA2 in the apo and ADP-bound state and the cryo-EM structure of ParA2 filaments assembled on non-specific DNA.

A number of ParA protein structures have been characterised and described in the literature. The solution of ParA2_{VC} allows some interesting comparison to be drawn and contributes to a better understanding of this family of Walker-type ATPases. The element of novelty of this work comes from the determination of the ParA2 filament structure that has not been previously described for any ParA protein. The details revealed by this structure shed new light on ParA dimer-dimer and ParA-DNA interactions. Overall, these findings contribute a new perspective to the field of genome segregation.

The storyline is clear and logical. Experiments appear to have been rigorously performed and include appropriate controls. The results support the conclusions. Figures are well organised and annotated. The videos are interesting and help to a large extent to visualize the three-dimensional details of the structures.

I have some queries about some experiments and minor suggestions to improve the manuscript.

1. Figure S5, EM of ParA2 in the presence of DNA and nucleotides: the filaments observed in the presence of ATP or ADP or ATP- γ -S appear quite different. The ParA2-ATP filaments appear dark positively stained, whereas the filamentous structures visible in the ParA2-ADP panel are negatively stained and the ParA2-ATP- γ -S are again quite different. Can the authors clarify and rationalize these differences?

We are grateful for this reviewer's very positive comments. The difference in staining was mostly due to poor grid quality, so we have included better micrographs for the ATP-bound and ADP-bound states, which we think illustrate more effectively the difference in the filament nature between these states: straight and well-ordered when bound to ATP, poorly ordered and lacking clear patterns when bound to ADP. We have also clarified this in the text (lines 239-241 of the revised manuscript). In the ATP γ S sample, we used Salmon Sperm DNA instead of plasmid, which is why they appear as a short line instead of a circular filament (See materials and methods).

2. The authors state that ParA2-DNA filaments could only be obtained at high protein concentration, as they dissociate at lower protein concentration. I am curious as to why ParA2-ADP-DNA filaments would be unstable, as in this scenario there is no hydrolysis. I guess there are other factors affecting the stability.

Our hypothesis is that the filaments are stabilised in the presence of tri-phosphate nucleotides, which is why they are only highly stable with ATP γ S. When bound to ADP the ParA2_{vc} filaments are not stable as we show in the negative stain. With ATP we see filaments at high

concentration, however, when we dilute the sample ATP hydrolysis causes for the third phosphate to depart and the filament to dissociate. Further experimental evidence would be required to verify this hypothesis. We have clarified this in the manuscript (lines 253-254).

3. Have the authors tried to perform these EM experiments using a ParA2 mutant impaired in ATP binding? If the DNA binding requires ATP binding as a prerequisite, then one would expect to observe no filaments. Additionally, the observation of an ATPase defective ParA2 mutant would provide support to the hypothesis that ATP hydrolysis triggers the ParA2-DNA filament disassembly.

A previous study (Fung, Bouet and Funnell., 2001) had identified the mutation K122R in P1 ParA (of P plasmid *E.coli*) which reduces its ATPase activity. We therefore engineered several mutations in the equivalent residue in ParA2_{vc} (K124) to R, Q and E. We have included in the revised manuscript negative stain EM for these mutants in the presence of ATP and DNA to view their effect on filament formation (figure S5b). We observe that the conservative K124R and K124Q mutants form stable filaments, consistent with reduced ATPase activity. In contrast, there is no filament formation observed with the K124E mutant, suggesting that this mutant is unable to bind to nucleotide and therefore form stable filaments with DNA. Further biochemical characterisation of these mutants will be published elsewhere. This is updated in the revised manuscript (lines 244-251).

4. Abstract: line 20, 'parS' should be italicized.
5. Introduction: line 37, 'The Par loci' should be 'The par loci'.
6. For consistency, use always upper or lower case when describing type I or type II and III systems throughout the manuscript.
7. Introduction: line 44, the system does not encode the centromere site; I would suggest to change the sentence to 'The type I segregation system locus encodes the ATPase ParA; an adapter protein, ParB; and contains centromere-like parS site(s)'.
8. Introduction: line 52, change to 'ParA forms...'
9. Introduction: line 57, delete 'in'
10. Introduction: line 82, upper case for Gram-negative.
11. Introduction: line 85, delete 'having'
12. Introduction: line 92, no need for upper case for 'Chromosome 2'.
13. Results: line 114, rephrase the sentence, P1 should not be Italic.
14. Results: line 116, I would suggest to change the sentence to 'to characterize ParA2_{vc} structure, to verify if it adopts a similar architecture...'
15. Results: line 153, it should be P7 rather than T7.
16. Results: line 158, no need of period at the end of the heading here and elsewhere.
17. Results: line 251, Italic for 'Helicobacter pylori'.
18. Results: line 265, upper case for 'c-terminus'.
19. Results: line 295, the sentence can be better worded, for example 'Collectively, this evidence supports the hypothesis that the oligomerization interface is conserved across ParA proteins encoded by chromosomes and plasmids'.
20. Discussion: lines 334-5, 'When encountering a DNA cargo bound to DNA'...this sentence is not clear, it needs clarification.

21. Discussion: line 335, 'leading to disassembly from the DNA'.
22. Discussion: from line 367 to 376, a different referencing style is adopted and two references are cited using author names. This needs to be rectified.

We thank the reviewer for carefully going through the manuscript and highlighting these errors, we have amended the typos and made the small changes suggested in the revised manuscript.

Reviewer #2 (Remarks to the Author):

This manuscript "The cryo-EM structure of the bacterial 1 type I DNA segregation ATPase filament reveals its conformational plasticity upon DNA binding.", by Parker et al., investigates the structural properties of the partitioning protein ParA from the chromosome 2 of *Vibrio cholera* (ParA2). It reports the crystal structure of ParA2 in several nucleotide bound states, and the structure of ParA2-ATPgS bound to DNA by cryo-EM at high resolution (4.5 angstrom). They build an atomic model of ParA2-DNA filament using the ParA2-ADP crystal structure. It reveals large conformational changes in the ATPgS and DNA bound form allowing the authors to provide a mechanistic model for the cooperative DNA binding of ParA.

The manuscript is well written and data are clearly presented. The data presented are largely confirmatory to what is already known for type Ia ParA in general and ParA2 in particular. The reported structures are highly similar to those of ParA from plasmids P1/P7 to which ParA2 is a close homolog. The filament formation on DNA, in a left-handed helical conformation, was also described earlier. However, authors applied cryo-EM technique, which provide with a very resolution allowing solving the structure of ParA2 in the filament form. It provides the numerous ParA-DNA contacts. They also detect a large conformational change involving the helix 1 and 2. They proposed that this conformational rearrangement explain the cooperativity of ParA on non-specific DNA, and based on amino-acids conservation that this property may be share amongst other type Ia ParA. This structural changes in the nucleotide bound form associated with DNA binding allow the formation of a charged surface that permits electrostatic interactions for adjacent dimers. The authors did not know whether this rearrangement is due to ATPgS binding, DNA binding or both.

Lastly, they discussed that the filaments formed by ParA2 and MinD (a member of the ParA/MinD superfamily) display completely different architecture and used different interfaces for dimer-dimer contacts, thus that the filament formation by these two distinct subfamilies are evolutionary unrelated.

This study is well performed and report the structure of ParA2 in its "filament" conformation. It also provided an explanation, based on a large conformation change, for the DNA binding cooperativity. While this is very interesting, most of the data are confirmatory and best suited in a more specialized journal.

The Table S1 is provided without title and legend.

We are grateful for this reviewer's positive comments and for highlighting this oversight, we have added a title and legend to Table S1.

Minor comments:

Introduction

§1: "motor protein" is not defined; are these true motor proteins as in Euk or motor-like based on activity or mode of action; "motor" has to be defined.

We have removed this term from our introduction and replaced with NTPase.

§2: ParB also display sequence independent DNA binding activity, as shown for P1/P7 but also for chromosomal ParB. This should be corrected.

We have corrected this in the revised manuscript (line 48) and added a corresponding reference.

§2: "ParB stimulates Par's ATPase activity". Some mechanistic details have been shown for this stimulation through arginine-finger like motif. At least, references for this more recent mechanistic detail should be included.

We have added this detail to the introduction in the revised manuscript (line 52).

§5: "in type Ib systems, parS is located after the promoter, followed by parA and parB." Is this true for all Ib type? Some reference would be welcome.

We have added some clarification to this sentence and added a corresponding reference (lines 85-86).

§7: The size/unit of Chr1 and Chr2 has to be corrected.

We have fixed this typo in the revised manuscript.

Throughout the manuscript, for stating that no filament formation has been revealed in vivo, it would gain in precision to mention and reference explicitly that experiments has been performed by super resolution microscopy instead of "fluorescence microscopy" or "fluorescence imaging" since these latter does not have the resolution capability for this argument.

We have changed the manuscript accordingly.

Results

p. 8, last §: "they dissociate at lower protein concentration (Figure S5): This is not shown directly in the Figure. At least, the figure legend should be completed by indicating what experimental conditions have been tested and describe the results.

We have clarified in the text the approximate ratio at which we observe filaments, (lines 237-239). We did not include in figure S5 the micrographs of diluted samples because they are just empty grids. We emphasise that negative stain EM is not a quantitative method, and therefore

further biochemical measurements would be necessary to establish the exact concentrations required for filament formation which is beyond the scope of this study.

p. 11: "the basic residues mentioned above are largely not conserved across ParA orthologues". These basic residues (K44, K74, H79) are in the Nterm domain, which is not present in type Ib ParAs. They could therefore not be considered conserved or not. This sentence should be clarified.

The basic residues mentioned in the text also include those in the C-terminus (K326, K327, R350, R352, K376, K377) that are not part of the N-terminal domain of type Ia ParAs. We have amended the text to clarify this and added that there is lacking of conservation of these N-terminal residues in respect to their type Ia orthologues specifically (line 295-297).

Reviewer #3 (Remarks to the Author):

This manuscript presents X-ray and cryo-EM studies of *Vibrio cholerae* ParA. The crystal structures reported captured various nucleotide states of ParA dimer, while cryo-EM of the ParA filament, in the presence of DNA, is the first near-atomic resolution structure of the ParA filament. An earlier structure for the ParAvc filament was reported by Hui et al in 2010 (PNAS), but that was done at very low resolution in negative stain. The authors are able to show that the ParA dimer is stabilized by nucleotide binding and observe a dramatic structural rearrangement upon DNA binding and filament assembly. Below are some comments and suggestions that need to be addressed in a revised manuscript.

Major point:

(1) My biggest concern is the model quality for both X-ray models and cryo-EM models. This can be easily judged from the statistics table. For crystal structures, extremely low Ramachandran favored (%), 83% and 89%, are reported. These are even worse than most recent cryo-EM models. Both models also have unreasonably high atom B-factors (85 and 124). Finally, the high clashscore (44 and 15) and high Bond angle/length RMSD also confirm the models were poorly refined.

We thank the reviewer for their comments. We have further refined the ADP structure, which has improved Ramachandran, clash score and bond angles RMSD (see table 1). We were not able to improve B-factors and we think that this reflects the poor quality of the structure, which we discuss in the reviewed text (lines 516-522).

Unfortunately, we were not able to refine further the APO structure, despite our efforts. This is because there is clear flaw in the diffraction data, unfortunately this appears to be reproducible as we collected multiple data sets but could not improve on our statistics. We comment on this as a feature of the APO- dimer which is inherently flexible, in the manuscript (lines 499-505).

(2) There are similar issues in the cryo-EM model: terrible Ramachandran, Rotamer, clashscore

statistics all indicate the model has been poorly refined. It is the authors' responsibility to properly refine a model.

We have further refined the cryo-EM atomic model, showing improved bond length, bond angle, Ramachandran, B-factors and clashscore and we hope that these statistics will be considered suitable considering the relatively low resolution of the map (see table 2).

(3) Only a map:map FSC is provided for the cryo-EM resolution estimation. This is often useless, as people have shown in many cases that this only measures the consistency between two half maps and can give a 4.0 Å resolution estimation even when the helical symmetry is wrong. A Model:map FSC must also be provided for resolution estimation.

We have included a map-to-model FSC in S6b. We emphasise that even at low resolution this map-to-model FSC only reaches a maximum of ~0.7, which is because there is no atomic model for the subunits at the edge of the filament as there is no full density for them (as seen on S7a). Consequently, the 0.5 cut off for this curve is not relevant for this structure.

Nonetheless, clearly the stated resolution matches the features of the map, with sidechains visible for helices in the central dimers (S7d-e), this demonstrates that we do not have the wrong helical symmetry.

Minor points

(4) A monomer docked into the 4.5Å cryo-EM map needs to be shown in the paper. The map should be adjusted to higher threshold than what is shown in Figure S7, so the readers can tell the quality of the map better.

We have replaced figure S7c with two different views of a monomer in the EM map and changed the contour level to allow for the helices to be visualised clearly.

(5) ATP-γ-S is referred to as a non-hydrolyzable analog. This is not true, and it is a slowly hydrolyzable analog.

(6) line 352: "In keep with this" should be "In keeping with this"

We have amended the typos and made the small changes suggested in the revised manuscript.

REVIEWERS' COMMENTS

Reviewer #1 (Remarks to the Author):

The authors have performed the experiments that were suggested in the first review round and included the results in the manuscript, thank you.

The revised EM micrographs are an improvement compared to the previous ones and clearly show that nucleotide binding by ParA is critical to allow filament formation. The data do confirm that the filaments are stabilised by tri-phosphate nucleotides, as virtually no filaments are visible in the presence of ADP.

However, statements such as 'We show that filament formation is controlled by nucleotide hydrolysis' (abstract) and 'ATP hydrolysis triggers disassembly' (line 222) are still very speculative and should be toned down. The ParA-K124 mutants represent a useful addition to the plot of the manuscript, but do not fully lend support to the hypothesis that ATP hydrolysis causes filament disassembly. What the mutant proteins show is that, in the presence of (presumably) reduced ATPase activity, filaments are visible in EM experiments. As the authors suggest, further characterisation of the mutants is necessary to draw definitive conclusions. Thus, my suggestion is to modify and tone down the statements about the mechanisms of filament disassembly.

Some minor edits:

1. Line 60, lower case for 'the partition complex...'
2. Line 66, upper case for 'Type I segregation...'
3. Line 69, upper case for 'Type Ia' after period
4. Line 71, '...two subtypes: in type Ia', lower case for 'in'
5. Line 82, lower case for 'however' after semicolon
6. Line 401, the reference style is still incorrect, use number in place of author's name.

Reviewer #2 (Remarks to the Author):

The authors have made most of my suggested revisions. However, contrary to what is mentioned in the rebuttal, the following point has not been addressed:

- §2: ParB also display sequence independent DNA binding activity, as shown for P1/P7 but also for chromosomal ParB. This should be corrected.

Author's reply: "We have corrected this in the revised manuscript (line 48) and added a corresponding reference".

The exact same sentence, with no reference, is still present in the manuscript: " ParA also binds to DNA in the presence of nucleotide, but unlike ParB this interaction is sequence-independent."

ParBs also display a significant and measurable sequence-independent DNA binding activity. I may suggest the following refs but others are available: Taylor et al 2015 NAR ; Ah-Seng et al 2009 JBC ; Surtees 2001 JBC ;

Other minor comments:

- The term "motor" is still present in the legend of Fig 1. Please correct.

- lane 96: the references for this sentence concern only ParA2. Please indicate references for ParA orthologs on ATPase and non-specific DNA binding activities.

- lanes 259-261: the second part of the sentence reads awkward relatively to the 1st part. Please check.

Reviewer #3 (Remarks to the Author):

I am satisfied with the changes.

Reviewer #1 (Remarks to the Author):

The authors have performed the experiments that were suggested in the first review round and included the results in the manuscript, thank you.

The revised EM micrographs are an improvement compared to the previous ones and clearly show that nucleotide binding by ParA is critical to allow filament formation. The data do confirm that the filaments are stabilised by tri-phosphate nucleotides, as virtually no filaments are visible in the presence of ADP.

However, statements such as 'We show that filament formation is controlled by nucleotide hydrolysis' (abstract) and 'ATP hydrolysis triggers disassembly' (line 222) are still very speculative and should be toned down. The ParA-K124 mutants represent a useful addition to the plot of the manuscript, but do not fully lend support to the hypothesis that ATP hydrolysis causes filament disassembly. What the mutant proteins show is that, in the presence of (presumably) reduced ATPase activity, filaments are visible in EM experiments. As the authors suggest, further characterisation of the mutants is necessary to draw definitive conclusions. Thus, my suggestion is to modify and tone down the statements about the mechanisms of filament disassembly.

We are grateful to this reviewer for their supportive comments. We have remove this statement from the abstract, and toned down the corresponding section in the results (lines 252-255 of the revised manuscript) to more clearly specify that our data does not show directly this disassembly process, which is a hypothesis based on our data but requires further validation.

Some minor edits:

1. Line 60, lower case for 'the partition complex...'
2. Line 66, upper case for 'Type I segregation...'
3. Line 69, upper case for 'Type Ia' after period
4. Line 71, '...two subtypes: in type Ia', lower case for 'in'
5. Line 82, lower case for 'however' after semicolon
6. Line 401, the reference style is still incorrect, use number in place of author's name.

We have corrected these.

Reviewer #2 (Remarks to the Author):

The authors have made most of my suggested revisions. However, contrary to

what is mentioned in the rebuttal, the following point has not been addressed:
- §2: ParB also display sequence independent DNA binding activity, as shown for P1/P7 but also for chromosomal ParB. This should be corrected.

Author's reply: "We have corrected this in the revised manuscript (line 48) and added a corresponding reference".

The exact same sentence, with no reference, is still present in the manuscript:" ParA also binds to DNA in the presence of nucleotide, but unlike ParB this interaction is sequence-independent."

ParBs also display a significant and measurable sequence-independent DNA binding activity. I may suggest the following refs but others are available: Taylor et al 2015 NAR ; Ah-Seng et al 2009 JBC ; Surtees 2001 JBC ;

We thank this reviewer for pointing out this mistake, this had been corrected in the track-changed version of the revised manuscript but was somehow lost in the non-track-changed version. We have added the corresponding references (line 125).

Other minor comments:

- The term "motor" is still present in the legend of Fig 1. Please correct.
- lane 96: the references for this sentence concern only ParA2. Please indicate references for ParA orthologs on ATPase and non-specific DNA binding activities.
- lanes 259-261: the second part of the sentence reads awkward relatively to the 1st part. Please check.

We have fixed these.

Reviewer #3 (Remarks to the Author):

I am satisfied with the changes.